# Designing, Development, and Evaluation of an Informatics Platform for Enhancing Treatment Adherence in Latent Tuberculosis Infection Patients: A Study Protocol

Rohitashwa Kumar [1], Manmohan Singhal [1,*], Devendra Kumar [2], Ashish Joshi [3] and KM Monirul Islam [4]

1 School of Pharmaceutical & Populations Health Informatics, Dehradun Information Technology University, Dehradun 248009, India
2 School of Pharmacy & Technology Management, SVKM's NMIMS (Deemed-to-Be) University, Mukesh Patel Technology Park, Shirpur 425405, India
3 School of Public Health, The University of Memphis, Memphis, TN 38152, USA
4 Division of Epidemiology, Medical College of Georgia, Augusta University, Augusta, GA 30912, USA
* Correspondence: manmohan.singhal@dituniversity.edu.in

**Abstract: Introduction:** Digital health interventions are gradually being incorporated into the management of tuberculosis to ensure treatment adherence, but only a small number of trials focusing on latent tuberculosis infection (LTBI) care have tested and evaluated them. It is anticipated that 170 million persons with LTBI may eventually develop active TB; thus, treatment of LTBI patients is an important aspect, along with ensuring treatment adherence. Digital platforms can be beneficial to ensure treatment adherence in LTBI patients, as various studies have shown the positive impact of digital interventions in improving patients' treatment adherence and treatment outcome. This study aims to explore the various available digital interventions worldwide for treatment adherence in LTBI patients and develop an informatics platform for enhancing treatment adherence in LTBI patients. **Methods:** This will be a quasi-experimental study divided into three phases. In the first phase, a scoping review method will be used to conduct a systematic literature review using the PRISMA tool to report on various digital interventions focused on treatment adherence in LTBI patients. In the second phase, a text message-based digital platform will be developed, and in the third phase of the study, an evaluation of the digital platform will be done using qualitative and quantitative questionnaires. The study will be conducted using a mixed-methods approach between January 2023 and December 2023. The sample size will be 162 participants, of whom 81 will be assigned to an intervention group and 81 will receive the usual care from the respective chest clinic as a control group. **Results:** A descriptive analysis of demographic variables and other variables will be done. Continuous variables will be described as mean $\pm$ standard deviation (M $\pm$ SD), medians (inter-quartile ranges) (M (IQR)), and medians (5th percentile to 95th percentile) (P5-P95). A two-sample independent T-test, the chi-square test, and the Mann-Whitney test will be used for comparisons between groups. Treatment success between control and intervention will be compared through a chi-square test. **Conclusions:** The key finding of the study will be an understanding of the efficiency of digital platforms for improving treatment adherence in latent TB patients in India.

**Keywords:** latent tuberculosis; latent TB; LTBI; digital health intervention; informatics; treatment adherence

## 1. Introduction

One of the top ten causes of death worldwide is tuberculosis, which is caused by a single infectious agent known as "*Mycobacterium tuberculosis*" and accounts for more than 10 million infections annually worldwide [1]. The highest risk of developing TB disease in individuals is within the first two years after their recovery from the first episode of infection; however, they can remain at risk for their lifetime [2]. In addition, latent tuberculosis infection (LTBI) is another challenge, which is a stage of the immune system's

determined response to Mycobacterium tuberculosis antigen stimulation without clinical evidence of active TB [3]. Approximately one-quarter of the worldwide population is infected with LTBI, and around 170 million people's LTBI will be turned into active TB in the future [1,4]. The frequency of LTBI among the world's 1.7 billion people in 2014 was 23%, according to mathematical modeling [5]. More than 55 million people are at high risk of developing TB disease due to the recent incidence of LTBI infection, which reached 0.8% of the global population in just two years [5], and the lifetime risk of reactivation for LTBI patients is between 5% and 10% [3]. Previous studies reported few risk factors for LTBI among different regions around the globe. A study conducted by Chen et al. in China found that people living with human immunodeficiency virus (PLHW) and blood donors with close contact with TB patients are prone to LTBI [6]. Other risk factors evident in different countries are a prior history of TB, household contact, HIV/AIDS, transplantation with immunosuppressant use, tumor necrosis factor-alpha blockers, chronic renal failure, hemodialysis, and the use of corticosteroids [7,8].

Therefore, it is crucial to treat LTBI patients, and successful management of LTBI can prevent progression to active TB and be advantageous for the person, the community, and the country. Although there are currently available treatments that can delay the onset of active TB by 60% to 90%, they are not practical and carry a risk of catastrophic and deadly side effects. [3]. Thus, detecting and treating LTBI could help reduce the pool of potential TB cases [3]. Due to the long duration of treatment and associated side effects, adherence to the treatment of LTBI and treatment completion are very important factors [4]. The defensive effectiveness of the LTBI treatment regimen is diminished by nonadherence to the prescribed regimen. Poor adherence to LTBI therapy may also raise the chance of getting TB and lead to the emergence of drug-resistant TB in the patient [4]. However, there are limitations to digital tools for LTBI care, and a lack of adherence tools may become a barrier to the scale-up of LTBI therapy services; thus, to support adherence, digital technologies could be applied [4].

Short Message Service (SMS) and Video Observed Treatment (VOT) for medication help in completing the complete course of treatment have been widely used as digital interventions in TB programs [9]. Some countries have explored similar interventions for LTBI management; however, the numbers are very few. It is evident that digital health connects and enables individuals to control their health and wellness, with the help of approachable and encouraging provider teams operating in digitally enabled care settings [10]. Additionally, digital health technologies, such as the use of digital tools and mobile, electronic, and wireless technologies in the health sector [11], act as a transformative force, particularly in low- and middle-income countries where mobile digital connectivity has attained a particularly high penetration [12,13]. Keeping that in view, TB prevention and care programs have included digital health interventions in practice, and they have been tested and assessed in field-level trials [14]. However, for efficient deployment, digital health solutions must consider the broader context of the health system and patient population [15].

Early detection, adherence to therapy, and results have all improved because of innovations in the management of tuberculosis employing digital technologies [16]. Although only a few evaluations that involve LTBI patients have been conducted [16,17], studies describing the impact of digital technologies on LTBI are insufficient. Digital technology has been shown to boost patient treatment completion, and patients are driven to self-administer their treatments without outside assistance [16]. Additionally, mobile applications show strong agreement with measurements, which can eliminate the need for in-person follow-up following test administration and improve the management of latent tuberculosis infection [18,19]. Mobile phone subscriptions are exceeding 100% globally [20], and individuals using the internet are at nearly 50% [21]. Text messages can therefore be considered a desirable and affordable intervention that might be ramped up further for LTBI therapy using currently available strategies for LTBI adherence [9,18]. In the context

of the WHO's End TB strategy and the search for digital innovations, there is a need to understand the use of digital technology for LTBI care and management.

## 2. Current Status of Research Studies

Only a few research studies have been done in the exploration of digital platforms for latent TB care at the international level. An implementation study by Lam et al. (2018) was done in New York City to understand the use of video technology in increasing treatment completion for patients with latent TB infection [16]. All patients were on 3-month isoniazid and rifapentine treatment (3HP). The 3HP treatment regimen was given to a total of 50 patients out of 71 eligibles, and 196 patients were on the Directly Observed Treatment Short Course (DOTS) as a control group. Higher treatment completion was observed with the DOTS than that previously seen with the clinic DOT among patients on 3HP. Johnston et al. (2018) conducted a randomized control trial in Canada at two sites to assess the effect of a two-way short message service on LTBI adherence [18]. It was observed that weekly two-way text messaging did not improve LTBI completion rates compared to standard LTBI care; however, completion rates were high in both treatment arms. The trial was unblinded except for the data analyst. A total of 358 participants were assigned to the intervention ($n = 170$) and control ($n = 188$) arms.

In the study of Holzschuh, EL et al. (2015) in Kansas, the use of video-directly observed therapy (VDOT) improved the treatment completion rates in 15 selected patients [22]. Out of 15, 14 patients completed the treatment. Moreover, van der Kop ML et al. (2014) are doing an RCT to understand the effect of weekly text-message communication on treatment completion among patients with latent tuberculosis infection in British Columbia, Canada [23]. Recruitment of 350 individuals initiating a 9-month isoniazid regimen has been done. To summarize, VDOT and text messages have been used in most studies, and both methods have shown improvement in treatment adherence. At the Indian level, there is a limitation of studies and a need for more studies in the field of latent TB management, which can help develop mechanisms to improve treatment adherence and completion rates [24].

## 3. Rationale of the Study

As per data from the Statista website, mobile phones, or cellular phones, are continuously expanding in India [25]. Digital Adherence Technologies (DATs) based on mobile phones or cellular phones may support alternate approaches for improving adherence [26]. Short messaging service (SMS) texts, digital pillboxes, and ingestible sensor technologies have been used so far under tuberculosis programs. DATs can support patients by reminding them to take the medication on a timely basis, observing pill taking digitally, and collecting dosage histories, which will be helpful in ensuring tailored care by TB programs [26]. Research studies will be useful to understand whether DATs are acceptable to healthcare providers and patients. It will also be useful for understanding their effectiveness in improving health system efficiency and treatment outcomes [26].

In India, as per the National Strategic Plan (NSP) 2020-25 of the Central TB Division, expanding TB preventative therapy is crucial to achieving the objectives of eliminating TB in India [27]. Thus, more focus has been put on treating LTBI patients. The guidelines for programmatic management of TB preventive treatment in India describe that both at the individual and group levels, LTBI course adherence and treatment completion are significant predictors of therapeutic improvement [28]. It further mentions that poor LTBI treatment compliance or early treatment discontinuation may raise the chance of developing TB, especially drug-resistant TB. There is a lack of adherence-related informatics platforms in India that support healthcare providers and patients to manage LTBI-related issues. Thus, the focus has shifted in this direction. The research study will be helpful for the development of an informatics platform that is effective in the context of treatment adherence for LTBI. Through this study, an informatics platform will be developed and evaluated for enhancing treatment adherence for LTBI.

## 4. Aims and Objectives of the Study

Aim: To develop, implement, and evaluate an informatics platform for enhancing treatment adherence for latent TB infection in India.

Objectives of the Study

Objective 1: Review the evidence on the informatics platform related to LTBI treatment adherence and identify the associated gaps, challenges, and opportunities.

Objective 2: Design and implement an informatics platform for enhancing treatment adherence for LTBI.

Objective 3: To pilot and evaluate the informatics platform for LTBI treatment adherence.

## 5. Geography, Methodology, and Study Design

This will be a quasi-experimental study done in government chest clinics in Delhi State, India, where treatment for latent TB is being provided. At the initial stage, one chest clinic will be selected to recruit the patients; however, if the sample size does not complete in one clinic, patients will be recruited from one additional chest clinic. The tentative period of recruitment will begin in January 2023 and last until the sample size is complete. Patients who have been prescribed six monthly isoniazid treatment regimens for LTBI treatment will be recruited. Eligible participants who show an interest in participating would be assigned to the intervention group or control group. At the initial level, patients' demographic data will be collected through a questionnaire. Written consent will be obtained from all participants who agreed to participate in the study. In the case of minor patients, consent will be obtained from the guardians or parents.

During recruitment, all participants will be oriented on the research study objectives, process, and roles. The chest clinic will help in the selection and recruitment of participants as and when they start the latent TB treatment of any new patient during the recruitment period. Ethical approval has been obtained from the University Research Ethics Committee of Dehradun Information Technology University, Uttarakhand, India, for this study.

### 5.1. Methodology and Study Design

The study will be conducted using a mixed-methods approach between January 2023 and December 2023. The qualitative study will be done through semi-structured, face-to-face, in-depth interviews with randomly selected LTBI patients and health service providers. For the quantitative component, data will be collected at baseline and end line through a questionnaire for both intervention and control groups for the outcome variables. The collected data will be used to assess the use and utility of the digital platform developed.

This study is divided into three phases (Figure 1):

Phase 1: In this phase, a scoping review method will be used to conduct a systematic literature review using the PRISMA tool to report on various digital interventions focused on treatment adherence in LTBI patients. The approach developed by Arksey and O'Malley [29]—which has since been updated by Levac et al. [30] and the Joanna Briggs Institute (JBI) [31]—will be used for the scoping review. Based on these approaches to scoping reviews, the following five stages or categories are identified: (a) defining the research issue; (b) finding pertinent studies; (c) choosing the appropriate studies; (d) synthesizing and evaluating the information that was retrieved; and (e) compiling, summarizing, and presenting the results.

Searches will be carried out in the PubMed, Web of Science, Scopus, Cochrane, and Embase databases for the period from 2016 to 2021. The reference list of all included articles found through this search will be scanned for any eligible literature. Language criteria will be applied, and articles published in the English language will be included. Randomized control trial (RCT), quasi-experimental study designs, and before and after study designs with or without a comparison group that used digital health interventions delivered by any means (mHealth, eHealth, SMS, etc.) based on LTBI will be included in this review.

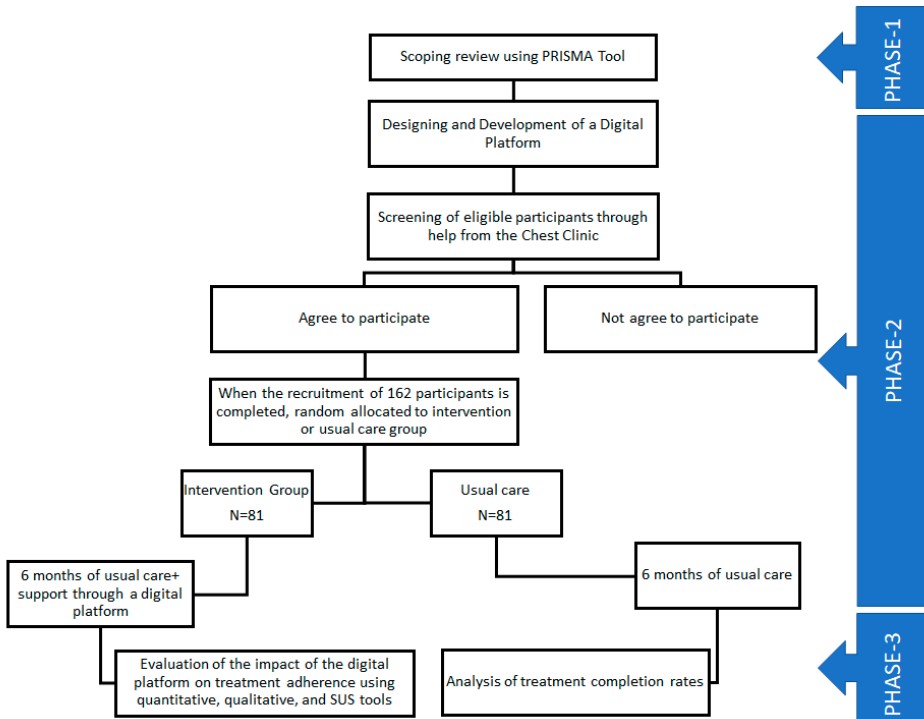

**Figure 1.** Phases of the study.

This review will be helpful in understanding the various modules of available informatics platforms for LTBI treatment adherence, along with their impact and gaps.

Phase 2: An informatics platform will be designed and implemented during the second phase, which can support enhancing treatment adherence for LTBI. A dashboard will be created that will be based on text message responses from the patients through the mobile application. A specifically designed mobile application will be given to intervention groups to report their daily medication and side effects. To participate in the study, patients from the intervention group will need to install the mobile application on their Android smartphones. This application will provide support to patients in reporting daily medication and side effects if they are experiencing any through the health update section of the application. The messages and notifications sent through the dashboard will be visible in the notification section of the application. Regular motivational messages will also be sent through the dashboard to motivate the patients and complete the treatment course. The dashboard will support health facilities in regular monitoring of the patient's treatment, and regular text messages will help patients get reminders for medicine. Globally, there are some research studies available that have shown significant improvement in the treatment adherence of latent TB patients through text message-based intervention [9,18]. However, linking this text message-based intervention with the dashboard has not been explored yet.

The dashboard will be designed with the help of an IT person, and messages will be sent daily to all patients. Patients' responses will be recorded on the dashboard, and the researcher, supervisor, and treating physician will have access to the patient's data. The language of notifications and messages sent through the dashboard will be Hindi.

The criteria for the inclusion and exclusion of patients are as follows:

Inclusion criteria

1. Patients who will start LTBI treatment;
2. Patients have mobile phones;
3. Patients who give written consent and are willing to participate;
4. Patients understand Hindi;
5. In the case of patients less than 18 years of age, parents are willing to participate.

Exclusion Criteria

1. Patients who do not have mobile phones;
2. Patients who do not understand Hindi;
3. Patients are not willing to sign a consent form;
4. Patients suffering from active tuberculosis.

Phase 3: During this phase, the informatics platform will be evaluated using qualitative and quantitative methods. Randomly chosen eight to ten LTBI patients and the medical officer of the chest clinic will participate in semi-structured, in-depth face-to-face interviews for the qualitative study. Data will be gathered for the outcome variables for both the intervention and control groups for the quantitative component using a questionnaire. The System Usability Scale (SUS), a tool that has been extensively tested throughout the world, will be used to gauge the utility and acceptability of the proposed digital intervention.

*5.2. Materials*

Participants will be told that they are taking part in the study and will be given a debriefing session. A consent form will be given to all participants to sign them up for participation in the study. As qualitative tools, a semi-structured interview checklist will be used, and questionnaires in hard copy will be used for baseline and end-line data collection for quantitative data collection.

*5.3. Target Population and Sample Size*

Household contacts of pulmonary TB patients regardless of age, who will start a treatment course of six months of daily isoniazid will be recruited. Convenient sampling will be done to reach the desired level of the sample. The tentative period of recruitment will be between January 2023 and March 2023. Using a quasi-experimental study design and assuming two independent samples (the intervention and control groups; a 1:1 ratio), alpha = 0.05, power = 80%, the expected full-adherence rate in the intervention group = 80%, and adherence in the control group = 60%, the difference in adherence between the two groups is 20%, and the required sample size is 162 (81 in the control group and 81 in the intervention group). For the qualitative component, a total of 8–10 in-depth interviews will be conducted among LTBI patients in the intervention group.

*5.4. Data Analysis and Results*

SPSS statistical software 25.0 will be used for statistical analysis. A descriptive analysis of demographic variables and other variables will be done. The Shapiro-Wilk test, histogram, and QQ graph will be combined to detect data normality. Continuous variables will be described as mean $\pm$ standard deviation (M $\pm$ SD), medians (inter-quartile ranges) (M (IQR)), and medians (5th percentile to 95th percentile) (P5–P95). Categorical variables will be described as frequencies and percentages. A two-sample independent T-test, the chi-square test, and the Mann-Whitney test will be used to compare groups. Logistic regression will be used to examine the differences among various groups. Further, treatment success will be measured by the percentage of individuals who complete the treatment. Treatment success between control and intervention will be compared through a chi-square test.

*5.5. Outcome Measures*

The first outcome of this study will be an understanding of the available informatics platforms for LTBI treatment adherence in India and around the world. Based on the understanding of various available digital interventions for LTBI patients, a new informatics platform will be developed, which will be the second outcome. Treatment success will be seen as a third outcome, and the expected improvement in a treatment success rate of 20 percent and the impact of demographic characteristics on patients' treatment adherence will be counted as additional important outcomes.

*5.6. Data Confidentiality and Privacy*

Data confidentiality and privacy will be ensured throughout this study following the Government of India's policy on data protection. All the patient information will be kept with the researcher, district TB officer, and other clinical staff only, and all the files will be password protected. Information will not be shared with any other person, and with the research team, it will be shared with password protection. Dates and files will be stored in a secure area to which only researchers have access, and the computers on which the data will be stored will also be password-protected.

*5.7. Limitations*

This research study has some limitations as it will be conducted in the context of India. Secondly, it will be limited to a sample size of 162. The other limitation of the study is that it will be limited to smartphone users and not to those with feature phones.

## 6. Conclusions

The scoping review will provide information on various digital platforms available worldwide for supporting treatment adherence in LTBI patients, along with their impact, usefulness, and key components. The key finding of the study will be an understanding of the efficiency of digital platforms for improving treatment adherence in latent TB patients in India.

**Author Contributions:** All authors have contributed to the design of the study, the development of the questionnaire, and the preparation of the manuscript and have approved it for publication. All authors have read and agreed to the published version of the manuscript.

**Funding:** This research received no external funding.

**Institutional Review Board Statement:** The study was conducted in accordance with the Declaration of Helsinki and approved by the Institutional Review Board (University Research Ethics Committee) of Dehradun Information Technology University, Uttarakhand, India. Protocol Number: DITU/UREC/2022/04/8 Date of approval: 12 May 2022.

**Informed Consent Statement:** Informed consent will be obtained from all subjects involved in the study.

**Data Availability Statement:** Data from completed analyses will be available from the corresponding author upon reasonable request.

**Acknowledgments:** The authors are the only contributors to this manuscript and are acknowledged.

**Conflicts of Interest:** The authors declare no conflict of interest.

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
