# Peer review of "Designing, Development, and Evaluation of an Informatics Platform for Enhancing Treatment Adherence in Latent Tuberculosis Infection Patients: A Study Protocol"

_biomedinformatics, doi:10.3390/biomedinformatics3010016_

Round 1
Reviewer 1 Report
In this work, the authors proposed a protocol to study and develop a computational tool to enhance the treatment adherence of Latent Tuberculosis patients. This work is a three-stage protocol, starting with a systematic review, followed by platform development, and finishing with an evaluation stage.
The most positive aspect of this text is its introduction. The authors presented evidence that their work is potentially innovative. Furthermore, they assessed the importance of their target area and how the usage of technologies enforces their proposal.
Apart from this, the text has a minimal contribution. The authors provided very little information regarding evaluation methods, statistical hypotheses, and protocols. The only clear protocol is the systematic review, which is the well-known PRISMA. The text organization and writing style also require much further work.
Author Response
We appreciate your positive comments about our proposal. We also understand your concern about evaluation methods, statistical hypotheses, and protocol.
Our response to each of these items is mentioned below:
Evaluation methods and statistical hypothesis: We proposed a mixed-methods design including a systematic literature review, a quasi-experimental, and a qualitative method using semi-structured face-to-face interviews with randomly selected patients. Given the nature of the study, we decided not to have a statistical hypothesis. However, we proposed a rigorous statistical analysis to evaluate the outcomes of the study in the Study Design Section-5.1; pages 4-5.
We will also use the SUS (System Usability Scale) tool to understand the effectiveness and usability of the digital platform. We have added that now in the paper.
Protocol: We describe the study implementation protocol in the Study Design Section-5.1; pages 4-5.
Editing: we have done the proofreading of the paper and all errors have been rectified now. We have also edited the content wherever required.
Looking forward to your acceptance.
Thanks and regards
Rohitashwa

Reviewer 2 Report
The manuscript presents a study protocol to evaluate a text message based digitial platform for enhancing treatment adherence. Overall, this study method is well organized and the manuscript is well written, however, considering the population of India, the sample size is only 162 is too limited. I recommend the authors recruit more participaites to improve the integrity of the article
Author Response
Thank you for your appreciating the paper. We have described the sample size calculation and requirement using standard statistical parameters. You can find it in section 5.3. Target population and Sample size: Using a quasi-experimental study design and assuming two independent samples (intervention and control group; 1:1 ratio), alpha=0.05, power=80%, expected full-adherence rate in intervention group=80%, and adherence in control group=60%, the difference in adherence between two groups is 20%, the required sample size is 162 (81 in control and 81 in intervention group). For the qualitative component- A total of 8-10 in-depth interviews will be conducted among LTBI patients from the intervention group.
You are correct that increasing the sample size will give us better power in our statistical analysis. However, considering resources, time limitations, and approval from the ethical committee we are not in a position to increase the sample size now.
Editing: We have rectified the grammatical and punctuation errors in the paper and tried to improve the content and writing style.
Looking forward to your acceptance.
Thanks and regards
Rohitashwa

Reviewer 3 Report
The study protocol submitted by Kumar et al reports the use of digital technology in latent tuberculosis infection care. This manuscript is well structured overall, but it needs minor modifications, that I have outlined in the comments below.
1. Abstract is missing in the manuscript
2. “Mycobacterium tuberculosis” should be in italic font.
3. The authors have to explain how they are going to monitor the patient with multichroic condition?
4. Do the authors think monitoring the patient using digital platform can potentially help current LTBI practice? If so, why it is not successful with significant patient numbers?
5. What is the potential cost of the technology in LTBI treatment? Who will contribute?
6. Please show the flow diagram with tentative study selection process?
7. There are fifteen articles describing the use of digital technology in latent tuberculosis, please add them in your protocol.
8. What are the authors thought on emphasizing LTBI treatment should focus on individual at higher risk of developing active TB?
9. The authors have to proofread the manuscript, there are grammatical errors.
Author Response
Please find below the responses on the queries:
- Abstract is missing in the manuscript.
Response: we have added abstract
- “Mycobacterium tuberculosis” should be in italic font.
Response: we have edited the words in italics throughout the manuscript
- The authors have to explain how they are going to monitor the patient with a multichroic condition.
Response: we are not sure about this comment. We assume that reviewer 3 meant the patient monitoring process for this study.
All patients are registered in the hospital register with demographic characteristics, and clinical and laboratory information. It also collects the comorbidity conditions of patients. We will monitor all patients in a similar process. Also, we will adjust the comorbidity at the analysis stage to have an unbiased outcome measurement.
- Do the authors think monitoring the patient using a digital platform can potentially help current LTBI practice? If so, why it is not successful with significant patient numbers?
Response: As per the strategic plan of the National TB Elimination Program of India, the focus is being given to treating LTBI patients. The guidelines released by the National program mention that digital interventions are required to ensure treatment adherence in LTBI patients. Till now, for LTBI no specific digital interventions are in practice thus this platform will provide an option if successful. The major challenge in treating LTBI patients is an asymptomatic condition and after starting treatment they will require regular monitoring and motivation which can be possible through a digital platform. This platform will provide an option for treating physicians to look at the progress of treatment adherence in patients and help them in case of any side effects.
During scoping review detailed analysis is being done for various platforms available throughout the world to understand the various challenges of these types of platforms. The one challenge that was common is not using the local language for text messages and we have taken care of that aspect.
- What is the potential cost of the technology in LTBI treatment? Who will contribute?
Response: The potential cost of developing the platform will be approximately 2000 USD including server costs. The complete contribution is being done by First Author as it is not funded by other agencies.
- Please show the flow diagram with the tentative study selection process.
Response: We have added a flowchart on page no. 5
- There are fifteen articles describing the use of digital technology in latent tuberculosis, please add them in your protocol.
Response: We have tried including the most relevant articles in the study protocol and it would be great if you can please provide more references in this direction.
- What are the authors thought on emphasizing LTBI treatment should focus on individual at higher risk of developing active TB?
Response: In developed countries, all LTBI cases are treated with a short course of anti-TB drugs. Treating high-risk patients will bring the incidence and prevalence down. But the question remained unanswered, is it feasible in high-burden countries such as India?
- The authors have to proofread the manuscript, there are grammatical errors.
Response: We have reviewed the paper for English grammar and rectified the errors.
We hope our responses are fine on your end. Looking forward to your acceptance.
Thanks and regards
Rohitashwa

Round 2
Reviewer 1 Report
The style of the text has slightly improved, but it still requires more work (For instance, second paragraph in Aim 2). I keep the other critics, as the authors chose to present statistical tools without raising hypotheses.
Author Response
Dear Reviewer,
Thank you so much for the detailed and very important comments. We have tried our best to address all your valuable comments for improving the manuscript. We hope revised version is looking fine to you.
Looking forward to your acceptance.
Thanks and regards
Rohitashwa Kumar

Reviewer 2 Report
Accept in present form
Author Response
Dear Reviewer,
Thank you so much for accepting the revised manuscript. We appreciate your time given to this review and providing inputs about improvement areas in the manuscript.
Thanks and regards
Rohitashwa Kumar

Round 3
Reviewer 1 Report
Same as the last issue.